# Evaluation of Small-Molecule Candidates as Modulators of M-Type K^+^ Currents: Impacts on Current Amplitude, Gating, and Voltage-Dependent Hysteresis

**DOI:** 10.3390/ijms26041504

**Published:** 2025-02-11

**Authors:** Te-Ling Lu, Rasa Liutkevičienė, Vita Rovite, Zi-Han Gao, Sheng-Nan Wu

**Affiliations:** 1Department of Pharmacy, China Medical University, Taichung 406040, Taiwan; lutl@mail.cmu.edu.tw; 2Neuroscience Institute, Medical Academy, Lithuanian University of Health Sciences, Eiveniu 2, 50161 Kaunas, Lithuania; rasa.liutkeviciene@lsmuni.lt; 3Latvian Biomedical Research and Study Centre (BMC), LV-1067 Riga, Latvia; vita.rovita@biomed.lu.lv; 4Institute of Basic Medical Sciences, College of Medical, National Cheng Kung University, Tainan City 701401, Taiwan; s58131110@gs.ncku.edu.tw; 5Department of Research and Education, An Nan Hospital, China Medical University, Tainan City 709204, Taiwan; 6School of Medicine, National Sun Yat-sen University, Kaohsiung 804201, Taiwan

**Keywords:** M-type K^+^ current, M-type (K_V_7x or *KCNQx*) channel, current kinetics, small-molecule modulator

## Abstract

The core subunits of the K_V_7.2, K_V_7.3, and K_V_7.5 channels, encoded by the *KCNQ2*, *KCNQ3*, and *KCNQ5* genes, are expressed across various cell types and play a key role in generating the M-type K^+^ current (*I*_K(M)_). This current is characterized by an activation threshold at low voltages and displays slow activation and deactivation kinetics. Variations in the amplitude and gating kinetics of *I*_K(M)_ can significantly influence membrane excitability. Notably, *I*_K(M)_ demonstrates distinct voltage-dependent hysteresis when subjected to prolonged isosceles-triangular ramp pulses. In this review, we explore various small-molecule modulators that can either inhibit or enhance the amplitude of *I*_K(M)_, along with their perturbations on its gating kinetics and voltage-dependent hysteresis. The inhibitors of *I*_K(M)_ highlighted here include bisoprolol, brivaracetam, cannabidiol, nalbuphine, phenobarbital, and remdesivir. Conversely, compounds such as flupirtine, kynurenic acid, naringenin, QO-58, and solifenacin have been shown to enhance *I*_K(M)_. These modulators show potential as pharmacological or therapeutic strategies for treating certain disorders linked to gain-of-function or loss-of-function mutations in M-type K^+^ (K_V_7x or *KCNQx*) channels.

## 1. Introduction

The *KCNQ2*, *KCNQ3*, and *KCNQ5* genes are known to encode the core subunits of the K_V_7.2, K_V_7.3, and K_V_7.5 K^+^ channels, respectively. These channels are expressed across a wide range of excitable and non-excitable cells [1,2,3]. Upon activation by appropriate membrane depolarizations, these K_V_ channels generate the macroscopic M-type K^+^ current (*I*_K(M)_), which is characterized by its low voltage activation threshold as well as slow activation and deactivation kinetics [4,5,6]. The low voltage activation threshold allows *I*_K(M)_ to become active at a membrane potential close to the resting potential of the cell. The designation “M” in *I*_K(M)_ reflects its modulation by muscarinic receptors, originally identified as being activated by acetylcholine [5,7,8,9,10,11].

Among these channels, K_V_7.2 and K_V_7.3 serve as the primary molecular components of *I*_K(M)_, a key regulator of neuronal excitability, including spike frequency adaptation and synaptic transmission [1,2,6,12,13,14,15]. However, in specific locations, other subunits may also contribute to M-like currents [6,16]. K_V_7.1 (*KCNQ1*) is expressed in the heart, particularly in atrial and ventricular myocytes, where it contributes to the slow delayed-rectified K^+^ current (*I*_Ks_) [17]. In contrast, K_V_7.4 (*KCNQ4*) is primarily involved in the auditory system, where it plays a key role in maintaining K^+^ balance within the cochlea [18]. K_V_7.5 (*KCNQ5*) channels play a critical role in regulating cellular electrical excitability and are essential for the proper functioning of the nervous system and other excitable tissues. The dysfunction of K_V_5 channels has been implicated in various neurological and muscular disorders [2,14]. The *I*_K(M)_ is particularly sensitive to inhibition by linopirdine and can be differentiated from *ether-à-go-go*-related (*erg*)-mediated K^+^ currents [5,19,20,21]. Recent studies have suggested that the magnitude of *I*_K(M)_ may regulate the availability of voltage-gated Na^+^ (Na_V_) channels during prolonged high-frequency firing of action potentials, thereby affecting reliable presynaptic spikes and synaptic transmission in a quantum-dependent fashion [1,14,22,23].

Modulation of *I*_K(M)_ has gained increasing recognition as a potential therapeutic approach for treating a variety of neurological disorders associated with either excessive neuronal activity or dysfunctional autonomic control [2,3,10,12,15,24,25,26,27,28]. Genetic variants in the *KCNQx* genes, including both gain-of-function and loss-of-function mutations, have been associated with various conditions such as social impairments, cognitive dysfunction, neuropathic pain, and epilepsy [24,25,26,27,29,30,31].

## 2. Biophysical Characteristics of *I*_K(M)_

As illustrated in Figure 1, pituitary GH_3_ cells were exposed to a high-K^+^, Ca^2+^-free solution containing 1 μM tetrodotoxin, while the measuring electrode was filled with a K^+^-enriched solution. Tetrodotoxin was included in the bathing medium to eliminate interference from other ion channels, including Na_V_ channels and Na^+^-activated K^+^ channels [32]. Using a high-K^+^ bathing solution offers the advantage of conveniently distinguishing between different types of voltage-gated K^+^ currents, including *erg*-mediated K^+^ currents. These *erg*-mediated currents may partially overlap with *I*_K(M)_ in various cell types [19,33]. During the recordings, the cell was held at a membrane potential of −50 mV, and a 1 − s depolarizing pulse to −10 mV was applied to evoke *I*_K(M)_. Under these experimental conditions, *I*_K(M)_ displayed a characteristic slow activation profile in response to sustained depolarization, with activation and deactivating time constants of approximately 100 and 30 ms, respectively. Given that the reversal potential for K^+^ ions is approximately 0 mV in this context, the resulting *I*_K(M)_ was an inward K^+^ current that activated gradually, promoting the influx of K^+^ ions into the cell [4,5,21,34]. Moreover, when the voltage returns to the original holding potential (which is −50 mV), a large deactivating tail inward current is also generated (Figure 1).

Figure 2 presents the steady-state activation curve of *I*_K(M)_ in GH_3_ cells, illustrating its relationship with membrane potential. The amplitude of *I*_K(M)_ was measured at the end of each 1 − s depolarizing step from a holding potential of −50 mV. Data points recorded at various membrane potentials were fitted using the Boltzmann equation (or Fermi-Dirac distribution) [35]. The Boltzmann equation is given by$$\frac{G}{G_{\max}}=\frac{1}{1+\exp\left[\frac{-(V-V_{1/2})qF}{RT}\right]}$$
where *G* represents the ionic conductance of *I*_K(M)_, calculated as *G* = *I*/(*V − E*_rev_), (with *I* and *V* denoting the current amplitude and membrane potential, respectively, and *E*_rev_ being the reversal potential for K^+^ ions), *G*_max_ is the maximal conductance of *I*_K(M)_, *V*_1/2_ is the voltage at which half-maximal activation occurs, *q* is the apparent gating charge in units of the elementary charge (*e*), *F* is Faraday’s constant, *R* is the universal gas constant, *T* is the absolute temperature, and F/RT equals 0.04 mV^−1^.

In this context, the apparent gating charge refers to the amount of electrical charge that moves across the membrane during the activation of ion channels in response to a voltage change [35,36,37]. This “gating” charge corresponds to the movement of charged particles, typically amino acid residues in the ion channel protein, which is necessary for the channel to open [36]. Under our experimental conditions, the steady-state activation curve of *I*_K(M)_ (Figure 2) was optimally fitted with an upward nonlinear sigmoidal function, yielding a half-maximal voltage (*V*_1/2_) of −23.4 mV and an effective gating charge (*q*) of 5.1 *e*.

The difference in free energy associated with the gating of *I*_K(M)_ at 0 mV (Δ*G*_0_) was calculated based on a 2-state gating model, which includes a closed (resting) state and an open state. According to the model, Δ*G*_0_ for *I*_K(M)_ activation at 0 mV can be expressed as *q* × *F* × *V*_1/2_, where *F* is Faraday’s constant [35,38]. Using the values for *q* and *V*_1/2_, the difference in free energy involved in the gating of *I*_K(M)_ at 0 mV (Δ*G*_0_) was thus estimated to be 11.5 kJ/mol (or 2.75 kcal/mol). The role of phosphosphoinositide metabolism in the gating of *I*_K(M)_ and its relationship to free energy [39,40] warrants further investigation.

Upon exposure to a prolonged upright triangular ramp voltage (V_ramp_), distinct forward and backward amplitudes of *I*_K(M)_ were observed, indicating the presence of nonlinear and non-equilibrium voltage-dependent hysteresis (Hys_(V)_) of *I*_K(M)_ (Figure 3) [35,37,41]. The Hys_(V)_ property of *I*_K(M)_ plays a critical role in modulating the overall behaviors of excitable cells, including pituitary GH_3_ lactotrophs. This phenomenon represents a unique and distinct shift in ion-channel gating, where the voltage sensitivity governing charge movement depends on the prior state of the K_M_ (K_V_7x or *KCNQx*) channel involved [35,37,41].

When the membrane potential of an excitable cell undergoes depolarization, specifically along the ascending limb of the upright isosceles-triangular V_ramp_, the current strength remains relatively small as indicated by the Hys_(V)_ loop (Figure 3). However, during membrane repolarization, namely along the descending limb of this double V_ramp_, the amplitude of *I*_K(M)_ increases significantly, resulting in a pronounced alteration in the membrane potential. Consequently, the influence of *I*_K(M)_ on the excitable membrane is more pronounced during repolarization than during depolarization [37]. Like perovskite solar cells (PSCs) [42,43,44,45]), the Hys_(V)_ behaviors may therefore occur across various ionic currents, including *I*_K(M)_. The Hys_(V)_ behavior is a major challenge in PSCs as it affects the accuracy and stability of efficiency measurements since it is manifested as a discrepancy between the forward (voltage increased from short circuit to open circuit) and the reverse scan (voltage decreased from open circuit to short circuit) [42,44,45,46,47]. However, in small cells, the primary effect of *I*_K(M)_ may be its modulation of the resting potential and the associated input resistance.

The activation and deactivation time courses of the *I*_K(M)_ occur with a time frame of tens to hundreds of milliseconds. When an agonist is applied, the activation of *I*_K(M)_ takes several seconds, and the recovery of *I*_K(M)_ following agonist removal can extend beyond a minute [40]. However, it is important to note that when a prolonged upright isosceles-triangular V_ramp_, lasting up to 2 s (Figure 3), is applied, the concentration of phosphatidylinositol 4,5-bisphosphate (PtdIns(4,5)P_2_)—which resides almost exclusively in the cytoplasmic leaflet of the plasma membrane—may be altered, potentially influencing Hys_(V)_ strength. This change occurs because the breakdown of PtdIns(4,5)P_2_ by phospholipase C happens within a few seconds [39,40]. However, after the agonist is removed, the resynthesis of PtdIns(4,5)P_2_ from phosptatidiylinositol (PtdIns) can take anywhere from one to several minutes, depending on the cell type.

Moreover, a recent study reported that compounds containing COOH groups can activate *I*_K(M)_, along with modifications on the activation curve of this current [48]. These COOH-containing compounds may activate *I*_K(M)_ through a lipoelectric mechanism, which involves the lipophilic compounds interacting at the interface between the lipid bilayer and the voltage sensor of the channel [48]. However, it remains unclear how these lipophilic compounds specifically influence the voltage sensor of the channel and how this interaction affects the strength of Hys_(V)_ in response to a double V_ramp_ stimulus.

## 3. Small-Molecule Modulators Targeting *I*_K(M)_

A variety of both natural and synthetic molecules have been demonstrated to play key roles in regulating *I*_K(M)_ activity across different cell types, as summarized in Table 1 and Table 2.

### 3.1. Small Molecules Known to Inhibit I_K(M)_ (Table 1)

#### 3.1.1. Bisoprolol (BIS, Concor^®^, Cardicor^®^, Zebeta^®^, 1-[4-[(2-Isopropoxyethoxy)methyl]phenoxy]-3-(isopropylamino)propan-2-ol))

BIS is recognized as an oral selective β_1_ adrenergic receptor blocker commonly used in the treatment of hypertension and heart-related conditions, such as atrial fibrillation, heart failure, and postural tachycardia syndrome [50,51]. Because of its lipophilic nature, it facilitates entry into brain tissue to produce regulatory actions on central neurons [52]. Earlier reports have shown that BIS could bind to β_1_-adrenergtic receptors inherently existing in brain areas including the pituitary gland and hippocampus [53,54]. A previous study also disclosed the effectiveness of BIS in increasing blood prolactin levels [55].

Notably, previous studies have demonstrated that when GH_3_ cells were exposed to BIS, the amplitude of *I*_K(M)_ in response to sustained depolarization was effectively suppressed with an IC_50_ value of 1.2 μM [56]. However, the BIS-induced inhibition of *I*_K(M)_ amplitude in these cells was not affected by the subsequent addition of isoproterenol or ractopamine but was attenuated by flupirtine or ivabradine. Isoproterenol and ractopamine are known to bind to and activate β-adrenergic receptors, while flupirtine and ivabradine can increase the amplitude of *I*_K(M)_ [56,57,58]. Furthermore, in the cell-attached current recordings, BIS decreased the open probability of K_M_ (K_7_x or *KCNQx*) channels, along with a significant reduction in the mean open time of the channel [56]. The exposure to BIS not only produced a decrease in the maximal open probability of K_M_ channels but also shifted the steady-state activation curve along the voltage axis to depolarized potentials by approximately 7 mV. However, minimal change in the gating charge of such an activation curve was demonstrated in BIS presence. Consequently, as cells were exposed to 1 μM BIS, the difference in free energy (ΔG_0_) required for activation of the K_M_ channel at 0 mV in GH_3_ cells was reduced from 1.25 to 0.79 kcal/mol [56].

Earlier findings have suggested that the magnitude of *I*_K(M)_ may influence the falling phase of bursting firing or spike after depolarization [2,13,59]. The falling phase of burst firing refers to the part of the action potential (or series of action potentials) where the membrane potential rapidly repolarizes or returns to a more negative value after the peak of the action potential. More recent research has shown that BIS can significantly affect the deactivation of *I*_K(M)_ in response to a downsloping V_ramp_, ranging from −10 to −50 mV, with varying durations [56]. As depicted in Figure 4, upon returning to −50 mV, a slower ramping rate of V_ramp_ led to a progressive exponential reduction in the peak amplitude of deactivating *I*_K(M)_, with an estimated time constant of 98 ± 8 ms (n = 11). However, when cells were exposed to 3 μM BIS, the peak amplitude of the current was significantly reduced in an exponential manner, with a time constant of 65 ± 7 ms (n = 11). These results indicate that, as the duration of the duration of the downsloping Vramp is increased, the amplitude of deactivating *I*_K(M)_ decreases exponentially. Furthermore, the presence of BIS resulted in a time-dependent reduction of *I*_K(M)_ [56]. Therefore, the BIS-induced block of *I*_K(M)_ is not instantaneous but develops gradually over time, when the channels are opened upon rapid membrane depolarization. Additionally, the BIS-induced inhibition of *I*_K(M)_ in GH_3_ cells does not seem to be solely dependent on binding to β-adrenergic receptors, although these receptors may exhibit constitutive activity [54].

#### 3.1.2. Brivaracetam (BRV, (2S)-2-[(4R)-2-Oxo-4-propylpyrrolidin-1-yl]butanamide)

BRV (Brivact^®^, Brivlera^®^), a chemical analog of levetiracetam, is an orally or intravenously bioavailable racetam derivative with anticonvulsant (antiepileptic) properties that has appeared in a growing number of research papers [60,61]. Notably, it has also been recognized to be efficacious in the treatment of epilepsy and status epilepticus [60,61,62,63].

BRV has been shown to reduce pain behavior in a murine model of neuropathic pain [60,64]. Additionally, this compound has demonstrated anti-neoplastic effects in glioma cells [65]. It has also been reported that BRV can influence the functional activity of neurons (e.g., hippocampal neurons) or endocrine cells (e.g., pituitary lactotrophs) by binding with high affinity to the synaptic vesicle protein2A (SV2A) [66,67,68]. SV2A is recognized as a critical broad marker for neuroendocrine cells and can be bound to anticonvulsants [66,69].

An earlier study has shown that the exposure to BRV in pituitary GH_3_ cells led to a concentration-dependent inhibition of *I*_K(M)_ with an IC_50_ value of 6.5 μM [70]. The activation time constant of *I*_K(M)_ was effectively increased during GH_3_-cell exposure to 10 μM BRV. However, BRV can also suppress the peak amplitude of *I*_Na_ with an IC_50_ value of 12.2 μM in GH_3_ cells. A leftward shift in the steady-state inactivation curve of *I*_Na_ was observed in the presence of BRV [70]. Under the inside-out current recordings, addition of BRV to the intracellular side of the excised patch enhanced the probability of BK_Ca_ channels that would be open, without affecting the single-channel conductance of the channel. These observations therefore suggest that, in addition to being a high-affinity ligand for SV2A [66,67,68], BRV is capable of perturbing the amplitude and kinetics of ionic currents, including *I*_K(M)_. This reveals a potential unintended effect on the functional activities of different excitable cells, such as synaptic transmission [14].

#### 3.1.3. Cannabidiol (CBD, 2-[(1R,6R)-3-Methyl-6-(prop-1-en-2-yl)cyclohex-2-en-1-yl]-5-pentylbenzeine-1,3-diol)

CBD is a non-psychoactive cannabinoid derived from the *Cannabis* plant, known for its potential therapeutic effects. CBD was previously reported to modulate the activity of μ- and δ-opioid receptors [71,72]. It is among over 100 cannabinoids present in the plant and has been demonstrated to be effective in treating various medical conditions, such as epilepsy, bipolar disorder, inflammation, and cancer [73,74,75].

A recent study by Liu et al. (2023) demonstrated that exposure to CBD led to a concentration-dependent reduction in the amplitude of *I*_K(M)_ in pituitary GH_3_ cells, with an IC_50_ value of 3.6 μM. CBD also caused a rightward shift in the steady-state activation curve of *I*_K(M)_ without affecting the gating charge of the curve. Notably, the inhibition of *I*_K(M)_ by CBD was not reversed by the subsequent addition of naloxone, an opioid receptor antagonist. However, the amplitude of *I*_K(M)_ in GH_3_ cells was effectively reduced in the presence of either thyrotropin releasing hormone (1 μM) or liraglutide (1 μM) [4,76]. The liraglutide-mediated inhibition of *I*_K(M)_ may result from its interaction with glucagon-like peptide-1 (GLP-1) receptors, which are expressed in pituitary cells, as liraglutide is a synthetic analog of GLP-1 [77].

Additionally, CBD was found to suppress the density of both activating and deactivating *I*_K(M)_ in response to pulse-train depolarizing stimuli ranging from −50 to −10 mV, as shown in Figure 5. These results suggest that CBD-induced inhibition of *I*_K(M)_ remains effective even under conditions of high-frequency pulse-train stimulation [76]. However, the presence of 10 μM CBD had no effect on *I*_Na_ in GH_3_ cells. Previous studies have demonstrated that a train of depolarizing pulses can significantly alter the magnitude of *I*_Na_, a current that decays exponentially over time [78,79,80,81]. Furthermore, it has been shown that the *I*_K(M)_ magnitude can regulate the availability of Na_V_ channels during prolonged high-frequency firing [81]. Taken together, these findings indicate that CBD exposure in GH_3_ cells reduces the magnitude of *I*_K(M)_ during pulse-train stimuli. As a result, the availability of Na_V_ channels during sustained high-frequency firing may be significantly diminished, potentially impairing reliable presynaptic spiking and reducing synaptic transmission at elevated frequencies [1,14,22,23].

Since naloxone, an opioid receptor antagonist, did not affect the CBD-induced reduction in *I*_K(M),_ this suggests that CBD’s effect on *I*_K(M)_ is not mediated through opioid receptor binding. These findings indicate that CBD likely exerts a direct and rapid influence on *I*_K(M)_, independent of interactions with the cannabinoid or opioid receptor [75,76]. If similar effects are observed under culture conditions or in vivo, they could potentially influence the firing behavior of action potentials in cells. However, whether CBD inhibits or stimulates *I*_K(M)_ remains a topic of debate [23,49]. It is possible that its effects vary depending on the specific subtypes of K_M_ (K_V_7x or *KCNQx*) channels expressed in different cell types.

#### 3.1.4. Nalbuphine (NAL, Nubain^®^, 17-[Cyclobutylmethyl]-4,5-epoxymorphinan-3,6,14-triol Hydrochloride)

NAL has been recognized as a moderate-efficacy partial agonist or antagonist of the μ-opioid receptor and as a high-efficacy partial agonist of the κ-opioid receptor with its low affinity for either the δ-opioid receptor or the σ receptor [82,83]. It can be used to balance anesthesia, for preoperative and postoperative analgesia, and for obstetrical analgesia [84,85].

A previous report has demonstrated the effectiveness of NAL in suppressing the amplitude of *I*_K(M)_ occurring in mHippoE-14 hippocampal neurons [86]. This inhibitory effect appears to be direct and independent of its binding to opioid receptors [85,87]. The IC_50_ value required for NAL-mediated inhibition of *I*_K(M)_ in mHippoE-14 neurons is estimated to be 5.7 μM. These results reflect that NAL exerts a concentration-dependent action on the suppression of *I*_K(M)_ in these cells.

As mHippoE-14 neurons were exposed to NAL, the amplitude of *I*_Na_ was also effectively suppressed, with an IC_50_ value of 1.9 μM. The presence of NAL not only inhibited the maximal conductance of peak *I*_Na_ but also shifted the steady-state inactivation curve to hyperpolarized potentials by 12 mV; consequently, the difference in free energy required for NAL-inhibited *I*_Na_ of the curve was altered. In contrast, the gating charge of the curve obtained between the absence and presence of 3 μM NAL did not differ significant. These results indicate that the NAL exposure can inhibit the peak amplitude of *I*_Na_ in a voltage-dependent manner in mHippoE-14 neurons [86]. In light of these observations, it is conceivable that the inhibitory effect of NAL on multiple ion currents (e.g., *I*_Na_ and *I*_K(M)_) tends to be a direct interaction between the drug and the channels themselves. Such an inhibition occurring within a clinically therapeutic range is not linked to agonistic or antagonistic effects on opioid receptors.

#### 3.1.5. Phenobarbital (PHB, Luminal Sodium^®^, 5-Ethyl-5-phenylbarbituric Acid)

PHB is a medication belonging to a group known as barbiturate. PHB has been recognized to be an anticonvulsant and a hypnotic because it can facilitate synaptic inhibition in the central nervous system by acting on the γ-aminobutyric acid (GABA) type A (GABA_A_) receptors [88,89,90]. GABA_A_ receptor is a ligand-gated chloride ion channel, which is the most common inhibitory channel in the brain. Pentobarbital also belongs to barbiturates and is similar to PHB. PHB has been disclosed to suppress neurogenic inflammation and exert neuroprotective and even anti-neoplastic activities by modifying membrane ion channels other than chloride channels [91,92].

Recently, it has been demonstrated that PHB can regulate the magnitude of multiple types of ionic currents, including *I*_K(M)_, residing in Neuro-2a neuroblastoma cells [79]. The magnitude of PHB-induced suppression in *I*_K(M)_ was not be reversed by the subsequent addition of flumazenil or chlorotoxin. Chlorotoxin, derived originally from scorpion venom, is a blocker of Cl^−^ channels (e.g., CIC-3 channel), while flumazenil is a benzodiazepine receptor antagonist. Therefore, the PHB-mediated inhibition of *I*_K(M)_ in Neuro-2a cells tends to be acute and may be independent of GABA_A_ receptor-mediated Cl^−^ currents [88]. Moreover, Neuro-2a-cell exposure to 100 or 300 μM PHB also decreased the amplitudes of activating and deactivating *I*_K(M)_ during 1 − s pulse train stimulation. These findings showed that a PHB-mediated block of *I*_K(M)_ remained efficacious upon high pulse-train stimulation. It is therefore conceivable that the availability of Na_V_ channels during high-frequency firing of action potentials in unclamped cells became further retarded during the exposure to PHB, despite its effectiveness in suppressing the *I*_Na_ amplitude directly [79]. However, since the reported PHB effects range between 100 and 300 μM, it remains to be determined whether its selective effects are directly related to its influence on *I*_K(M)_.

#### 3.1.6. Remdesivir (RDV, GS-5734, Veklury^®^, (S)-2-Ethyl-6-methyl-1-(2-propylthazol-4-yl)-4,5-dihydro-1H-pyrrolo[3,4-d]pyrimidin-7(6H)-one)

RDV (GS-5734), a broad-spectrum antiviral agent, is recognized as a mono-phosphoramidate prodrug of an adenosine analog that metabolizes into its active form GD-44524, which is a C-adenosine nucleoside analog [93]. This compound, a nucleotide-analog inhibitor of RNA-dependent RNA polymerase, is thought to be highly active against coronaviruses (CoVs), including MERS-Cov and SARS-CoV-2 [93,94,95,96].

It has been recognized as a promising antiviral drug against an array of RNA viruses, predominantly through the targeting of the viral RNA dependent RNA polymerase [93,96,97]. Recent reports have also shown the effectiveness of RDV in treating patients with certain types of panencephalitis [98,99,100].

As shown in Figure 6, previous studies have demonstrated that RDV can inhibit the amplitude of *I*_K(M)_ in GH_3_ cells, with an IC_50_ value of 2.5 μM [101]. Cell exposure to RDV, using a long-lasting triangular V_ramp_, noticeably suppressed the strength of Hys_(V)_ for *I*_K(M)_ elicitation. Therefore, it is likely that RDV exerts a perturbing effect on this non-equilibrium property in K_M_ (K_V_7x or KCNQx) channels within excitable membranes [97,101].

Besides its inhibitory effect on *I*_K(M)_, RDV can inhibit the amplitude of *I*_K(DR)_ along with the increased rate of current inactivation during prolonged membrane depolarizations [101]. The RDV molecules tend to accelerate *I*_K(DR)_ inactivation in a concentration- and state-dependent fashion, implying that they reach the blocking site of the channel, only when the channel involved resides in the open conformational state. The value of the dissociation constant (*K*_D_) required for an RDV-induced block of *I*_K(DR)_ in GH_3_ cells was 3.0 μM. The EC_50_ value of RDV against SAR-CoV-2 residing in Vero E6 cells was reported to be 1.76 μM [95,102]. Therefore, RDV-induced inhibition of *I*_K(M)_, *I*_K(DR)_ or *erg*-mediated K^+^ current is more than likely achieved in vivo [97,101,103]. Therefore, RDV is not classified as a prodrug, and its inhibition of these K^+^ currents [101,102] seems to occur independently of its possible effects on RNA polymerase activity [94,103,104]. It needs to be mentioned that the administrated of RDV in patients with COVID-19 infection may cause cardiac adverse events, possibly due its perturbations on K_V_ channels in heart cells [105,106].

### 3.2. Small Molecules Known to Stimulate I_K(M)_ (Table 2)

#### 3.2.1. Flupirtine (FLU, Katadolon^®^, 3-(4-Fluorophenyl-2-methylamino-1-pyridin-1-yl-propan-1-one)

FLU belongs to a class of triaminopyridines and is a centrally acting nonopioid analgesic agent with muscle relaxing properties. It has been used for a variety of neurological disorders involving neuronal overexcitability such as epilepsy and neuropathic pain [15,107]. In addition to its use in pain management, FLU was reported to display muscle relaxant and anticonvulsant activity. Notably, this compound was demonstrated to be beneficial in treating the patients with human prion diseases [108,109,110].

Previous research has shown that FLU enhances the activity of GABA_A_ receptors, which is associated with increased stimulation of K_V_7 channels [111]. It is also well established that FLU can bind to and activate K_V_7.2–7.5 (*KCNQ2–5*) channels [6,112]. However, a study conducted in motor neuronal NSC-34 cells revealed that FLU induces a time-, concentration-, and state-dependent reduction in delayed-rectifier K^+^ current (*I*_K(DR)_) elicited by prolonged depolarizing pulses, without affecting the activation kinetics of the current [32]. Notably, unlike *I*_K(M)_, this type of *I*_K(DR)_, such as K_V_3.1-encoded current, is characterized by an activation threshold at high voltages, rapid deactivation kinetics, and minimal influence on the resting membrane potential [101,113,114,115,116,117]. Additionally, earlier studies have shown that suppressing K_V_3.1, a key component of the *I*_K(DR)_ in high-spiking neurons, can reduce membrane excitability and consequently decrease neuronal firing [14,113,114,115,116,117]. This decrease in excitability is attributed to weakened resurgent K^+^ currents or repolarizing efficiency, which delay the repolarization of high-frequency action potentials and subsequently slow the recovery of *I*_Na_ [14,113,115,116]. The resurgent K^+^ current refers to a unique and relatively less common form of K^+^ current that can occur in certain excitable cells, including neurons and cardiac myocytes. Similar to the resurgent Na^+^ current, the resurgent K^+^ current is characterized by a brief transient outflow of K^+^ ions during the repolarization phase of an action potential, but the dynamics and mechanisms behind it are distinct.

Importantly, a detailed description of the *I*_K(DR)_ inactivation time course at varying FLU concentrations, combined with simulations using minimal binding model that yielded a *K*_D_ value of 8.9 μM, led the researchers to propose that FLU may act as a state-dependent blocker for *I*_K(DR)_ [57]. Furthermore, cell treatment with *N*-methyl-D-aspartate (NMDA), an NMDA receptor agonist, did not alter FLU-induced suppression of *I*_K(DR)_, suggesting that the inhibitory effect of FLU on *I*_K(DR)_ is independent of interactions with NMDA receptors. Therefore, the dual effects of *I*_K(M)_ stimulation and *I*_K(DR)_ inhibition induced by FLU may synergistically reduce motor neuron activity, particularly during high-frequency action potential firing, assuming similar outcomes occur in vivo [10,14,27,32,57,113,116].

#### 3.2.2. Kynurenic Acid ((2E)-2-Amino-3-carboxy-5,6,7,8-tetrahydrobenzo[b]pyridine-1 Carboxylic Acid) and Its Aminoalkylated Derivatives

Kynurenic acid (KYNA) is a naturally occurring produce of the normal metabolism of amino acid L-tryptophan that has been reported to inhibit *N*-methyl-D-aspartate receptor (NMDAR) and neuronal nicotinic α_7_ receptors [118,119]. This compound, together with L-kynurenine, is thought to be an endogenous metabolite of L-tryptophan known to block NMDAR, and it has been frequently demonstrated to exert neuroprotective or anticonvulsant properties in the brain [119]. Previous studies have disclosed the effectiveness of KYNA in ameliorating the magnitude of *KCNQ* gene-(*KCNQ*)- or *HCN* gene-(*HCN*)-encoded currents [120,121].

It has been recently demonstrated that in GH_3_ cells, KYNA or KYNA-A4, another aminoalkylated amide derivative, could produce a stimulatory effect on *I*_K(M)_ in a concentration-, voltage-, and state-dependent fashion [34]. The EC_50_ value required for KYNA or KYNA-A4-stimulated *I*_K(M)_ was yielded to be 18.1 or 6.4 μM, respectively. The presence of KYNA or KYNA-A4 shifted the relationship of normalized *I*_K(M)_-conductance and membrane potential to a more depolarized potential with no change in the gating charge of the current. The presence of KYNA was found to increase the probability of K_M_-channel openings in GH_3_ cells [56], with no clear change in single-channel conductance [34]. Under whole-cell current-clamp potential recordings, KYNA and KYNA-A4 were found to reduce the frequency of spontaneous action potentials in GH_3_ cells [34]. This reduction in action potential firing is likely attributed to the activation of *I*_K(M)_, rather than an effect on either NMDAR activity or aryl hydrocarbon receptors [56,120,122].

#### 3.2.3. Naringenin (NGEN, 4′,5,7-Trihydroxyflavan-4-one)

NGEN is a major dietary flavanone, a type of flavonoid commonly found in citrus fruits, and is known for its potential bioactive effects on human health. Several studies have reported that it and other structurally related compounds could produce anxiolytic and antinociceptive actions [123,124].

A previous study provides direct evidence that NGEN has a stimulatory action on *I*_K(M)_ in NSC motor neuron-like cells [59]. The IC_50_ value required for NGEN-mediated stimulation of *I*_K(M)_ was found to be 9.8 μM, which is significantly lower than the value reported for the inhibition of HERG channels [125]. When NGEN was added, it shifted the steady-state activation curve of *I*_K(M)_ conductance to a more hyperpolarized potential in NSC-34 cells [59]. The lack of effect by NGEN on the gating charge of K_M_ channels in NSC-34 cells led the investigators to propose that the stimulatory effect of NGEN on K_M_ channels in NSC-34 cells is not mediated through a direct interaction on the voltage sensor in the channel.

Furthermore, earlier studies have demonstrated that NGEN directly stimulates BK_Ca_ channels [59,126]. Consistent with previous findings in vascular myocytes [126], NGEN-induced enhancement of the *I*_K(Ca)_ amplitude in HEK293T cells transfected with α-*hSlo* was associated with an increased probability of BK_Ca_-channel openings [59]. Thus, it is plausible that NGEN, along with other structurally similar compounds, may interact at specific regions to modulate the activity of both BK_Ca_ channels and K_V_7-encoded channels [59]. Additionally, in a mathematical model of hippocampal pyramidal neuron [127], doubling the conductances of both K_M_- and BK_Ca_-channels to simulate the effect of NGEN (10 μM) led to a significant reduction in the bursting activity of action potentials in the modeled hippocampal neuron [14,59].

Because of its lipophilicity, NGEN was reported to transverse the blood–brain barrier and subsequently penetrate into different brain regions [128], although the brain concentrations of NGEN vary based on local extracellular milieu in and around membranes and synapses. Therefore, NGEN-induced actions on the stimulation of K_M_ and BK_Ca_ channels may combine to affect the functional activities, if both ion channels are functionally expressed in central neurons in vivo [14].

#### 3.2.4. QO-58 (5-(2,6-Dichloro-5-fluoropyridin-3-yl)-3-phenyl-2-(trifluoromethyl)-1H-pyrazolol[1,5-a]pyrimidin-7-one)

QO-58 has been previously demonstrated to be an opener of the *KCNQx* (K_V_7x) channel [129,130]. It has been reported that this compound could increase the pain threshold of neuropathic pain in a rat model, namely chronic constriction injury of the sciatic nerve [130]. QO-58 could also exercise anti-nociceptive action on inflammatory pain in rodents [131,132]. The ameliorating effects of this compound have been viewed to be closely linked to its activation of *KCNQ* (K_V_7) channels [130,131]. However, QO-40 (5-(chloromethyl)-3-(nathphalen-1-yl)-2-(trifluoromethyl)pyrazolo[1,5-a]pyrimidin-7(4H)-one), a compound structurally similar to QO-58, has been reported to stimulate the activity of large-conductance Ca^2+^-activated K^+^ (BK_Ca_) channels [133].

It has been shown that the presence of QO-58 can concentration-dependently increase the amplitudes of *I*_K(M)_ and Ca^2+^-activated K^+^ current (*I*_K(Ca)_) in pituitary GH_3_ cells with EC_50_ values of 3.1 and 4.2 μM, respectively [134]. Under GH_3_-cell exposure to QO-58, the steady-state activation curve of *I*_K(M)_ was shifted along the voltage axis to a hyperpolarized potential with no change in the gating charge, leading to changes in free energy of *I*_K(M)_ activation. The Hys_(V)_ strength of *I*_K(M)_ activated by triangular V_ramp_ measurably increased by the QO-58 presence. Furthermore, cell exposure to QO-58 enhanced the probability of BK_Ca_-channel openings as well as shifting the activation curve of the channel at stead state toward the less depolarized potential; however, neither the gating charge nor single-channel conductance of the channel was affected during its exposure.

Alternatively, based on molecular docking predictions, the interactions between the K_Ca_1.1 channel and QO-58, as well as between the KCNQ2 channel and QO-58, were clearly demonstrated [134]. These findings suggest that QO-58-mediated dual activation of K_M_ (K_V_7x or *KCNQx*) and BK_Ca_ channels observed in GH_3_ cells may hold pharmacological or therapeutic significance, provided that similar effects can be confirmed in vivo [135,136]. Notably, advances in computational biology have significantly enhanced the ability to predict the druggability of ion-channel gene products, including K_M_ (K_V_7x or *KCNQx*) channels [137,138,139].

#### 3.2.5. Solifenacin (SOL, Vesicare^®^, [(3R)-1-Azabicyclo[2,2,2]octan-3-yl](1S)-1-phenyl-3,4-dihydro-1H-isoquinoline-2-carboxylate)

SOL, a member of isoquinolines, has been viewed as an oral anticholinergic, namely a competitive muscarinic (M_1_ and M_3_) receptor antagonist. It is also an antispasmodic agent used to treat the symptoms of overactive bladder, neurogenic detrusor overactivity, or urinary incontinence [140,141]. because of muscarinic (M_2_ and M_3_) receptor antagonists that have anticholinergic effects such as relaxation of the detrusor muscle in urinary bladder [140,141,142,143].

An earlier study demonstrated that in GH_3_ cells, exposure to SOL resulted in a concentration-dependent increase in the amplitude of *I*_K(M)_ during prolonged membrane depolarization, with a concurrent shortening of the current’s activation time course [21]. The EC_50_ or *K*_D_ value of SOL-stimulated *I*_K(M)_ was calculated to be 0.34 or 0.55 μM, respectively. Exposure to SOL caused a leftward shift in the steady-state activation curve of *I*_K(M)_ in GH_3_ cells. While SOL increased the activity of single K_M_ (or *KCNQx*) channels, no change in the single-channel conductance was observed [21]. Additionally, the *I*_K(M)_ in hippocampal mHippoE-14 neurons was also subject to simulation by SOL. Although the precise ionic mechanism underlying SQL actions on K_M_ (or *KCNQx*) channels remains unresolved, these findings suggest a novel and non-canonical mechanism by which the SOL molecule interacts with the K_M_ channel to enhance the whole-cell *I*_K(M)_, ultimately reducing the firing frequency of spontaneous action potentials [21].

As demonstrated above in Figure 3, the Hys_(V)_ of *I*_K(M)_ evoked by the long isosceles-triangular V_ramp_ (i.e., the upsloping and downsloping ramp) was revealed in GH_3_ cells [101]. The adjustments of such Hys_(V)_ have been noticed to serve a role in fine-tuning the activity of ionic channels (e.g., K_M_ channels) to respond when they are virtually needed [41,70,101,144]. Moreover, as shown in Figure 7, previous findings have disclosed the effectiveness of SOL in increasing the hysteretic strength of *I*_K(M)_ associated with the voltage-dependent activation of instantaneous *I*_K(M)_ in response to isosceles-triangular V_ramp_ [101]. The SOL-mediated increase in *I*_K(M)_’s hysteretic area was suppressed by further addition of linopirdine, an inhibitor of *I*_K(M)_. Under such scenarios, it is possible that intrinsic changes in the voltage dependence of the voltage-sensing machinery in K_M_ (or *KCNQx*) channels, namely voltage-sensing domain relaxation [35,41], would be instantaneously and dynamically modulated during exposure to SOL. Whether the Hys_(V)_ strength of *I*_K(M)_ can be linked to the phosphoinositide metabolism in different cell types remains to be fully elucidated.

Several studies have shown that many compounds or drugs—such as cibenzoline, doxorubicin, and quinidine—exert their effects not by directly targeting cardiac muscarinic receptors but rather by modulating G protein-regulated inwardly rectifying K^+^ (GIRK) channels themselves [145,146,147]. It has been also reported that, during visceral contraction or glandular over-secretion, atropine, an antagonist of muscarinic receptors, does not demonstrate significant reversal effects on its own [148]. Therefore, whether certain compounds with anticholinergic activity, such as SOL, act indirectly on muscarinic receptors or on the activity of K_M_ (K_V_7x, *KCNQx*) channels themselves remains a notable subject that requires further in-depth investigation and research in the future.

Recent ECG tracings have highlighted variability in the RR interval, a key indicator of heart variability [149]. This variability is closely linked to the function of the autonomic nervous system, particularly the balance of sympathetic and parasympathetic nerve activity. Heart rate variability testing combined with posture change maneuvers (such as supine to standing transitions) allows for the assessment of autonomic nervous system variations and the effects of small molecular modulators targeting *I*_K(M)_ on autonomic function [150]. Therefore, investigating the regulation of the *I*_K(M)_ and its influence on autonomic functions as demonstrated previously [9,10,11,12,30,136,151], including heart rate variability, offers a compelling opportunity for in-depth analysis and research.

## 4. Conclusions

Experimental studies have demonstrated that a range of compounds or drugs can directly and dynamically influence the magnitude and gating properties of *I*_K(M)_, either through inhibition or stimulation. Table 1 and Table 2 provide the two-dimensional chemical structures of the compounds that inhibit or stimulate *I*_K(M)_, respectively. These alterations in *I*_K(M)_ can significantly affect cellular excitability, modifying firing frequencies and patterns across various cell types [6]. The slow activation and deactivation kinetics of *I*_K(M)_, with time constants of approximately 100 and 30 ms, respectively, are believed to play a crucial role in synaptic delays. This is especially evident during slow synaptic processes in chemical transmission, such as long-term facilitation and depression [2,6,14,15,81]. Moreover, the subcellular and molecular basis for both the slow deactivation and the residual activation of *I*_K(M)_ [39,40] are needed to be further explored.

It has been demonstrated that magnetoelectrical nanoparticles can modulate the firing activity, including the generation and propagation action potentials [139,152,153,154]. The hysteretic behavior induced by these nanoparticles under direct-current magnetic field highlights a potential mechanism for aberrant electrical activity. This effect may stem from phenomena such as magneto-Coulomb interactions or electroporation-induced currents, which could subsequently alter neural network structures and function [2,38,43,152,154,155,156,157,158,159,160].

However, it remains unclear whether magnetic fields combined with superferromagnetic nanoparticles, as described previously [161], influence the Hys_(V)_ behavior of *I*_K(M)_, the gating of other ionic currents, or currents generated by membrane electroporation in various cell types [38,43,152,157,160]. Addressing these uncertainties thus requires further research. Similarly, the combined application of transcranial magnetic stimulation and intravenously administered magnetic nanoparticles are important and warrant investigation for its potential impact on the Hys_(V)_ behavior of ionic channels, electroporation-induced currents, or both. Despite these open questions, this non-invasive therapeutic technique shows promise for treating psychiatric conditions such as major depressive disorder, obsessive-compulsive disorder, and autism [162].

Mutations in *KCNQx* genes, leading to either gain-of-function or loss-of-function, can disrupt the electrical activity of excitable cells, assuming these genes are functionally and bioactively expressed [25,30,151]. This study provides proof-of-concept evidence that small molecules can effectively modulate the amplitude, gating kinetics, and instantaneous Hys_(V)_ behavior of *I*_K(M)_ in electrically excitable membranes. Because these proteins are predominantly located on the cell surface membrane, rather than sequestered in the cytoplasm or nucleus, they are more accessible to externally administered compounds. Furthermore, *KCNQx*-encoded proteins are strongly implicated in various diseases, such as epileptic disorders [25]. Their significant druggability, including the potential for subunit-specific modulation, makes them attractive targets for pharmaceutical development [16,81,137,138]. While techniques like polymerase chain reaction (PCR) testing and Western blotting are reliable for detecting gene and protein expression, additional experimental validation using electrophysiological methods across various cell types is crucial to uncover their functional and bioactive effects, fully confirming their therapeutic potential.

## Figures and Tables

**Figure 1 ijms-26-01504-f001:**
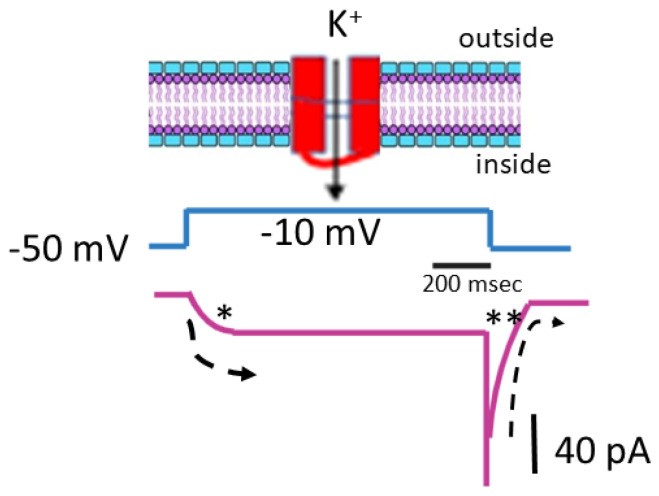
Biophysical characteristics of the M-type K^+^ current (*I*_K(M)_) present in pituitary GH_3_ cells. The cells were immersed in a high-K^+^ (145 mM) Ca^2+^-free solution with a reversal potential of around 0 mV, while we filled the recording pipette with a K^+^-enriched solution. The top diagram depicts a simplified schematic representation of a cell membrane (lipid bilayer) containing a K_M_ (K_V_7x or *KCNQx*) K^+^ channel. The solid arrow in the top diagram points to the activation of *I*_K(M)_ in response to membrane depolarization, with K^+^ ions moving inward. The middle diagram shows the voltage-clamp protocol (indicated in black), where the holding potential was set to −50 mV, and a depolarizing pulse was applied from −50 to −10 mV for a duration of 1 s. The bottom diagram represents a schematic representation of the *I*_K(M)_ trace (indicated in purple). The asterisk (*) marks the activation phase of the current, while the double asterisk (**) indicates the deactivation phase. The dashed curve illustrates the activation and deactivation time courses of *I*_K(M)_ in response to membrane depolarization.

**Figure 2 ijms-26-01504-f002:**
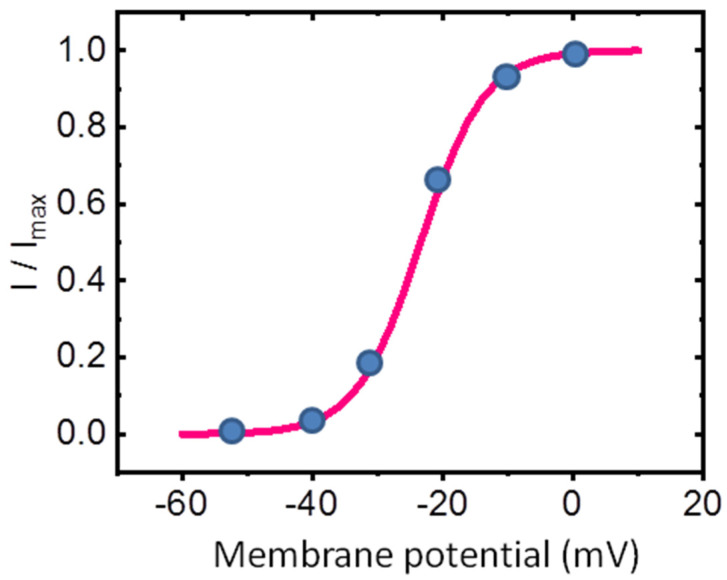
Steady-state activation curve of *I*_K(M)_ in GH_3_ cells. The activation curve of *I*_K(M)_ was derived from GH_3_ cells bathed in a high-K^+^ Ca^2+^-free solution. The relationship between membrane potential and normalized *I*_K(M)_ is depicted. The current amplitude was measured at the end of a 1 − s depolarizing pulse. The solid blue circles represent the normalized *I*_K(M)_ values recorded at each test voltage. The smooth sigmoidal red curve, generated by fitting the data points to the Boltzmann equation detailed in the text, highlights the voltage-dependent activation of *I*_K(M)_, particularly at low voltages.

**Figure 3 ijms-26-01504-f003:**
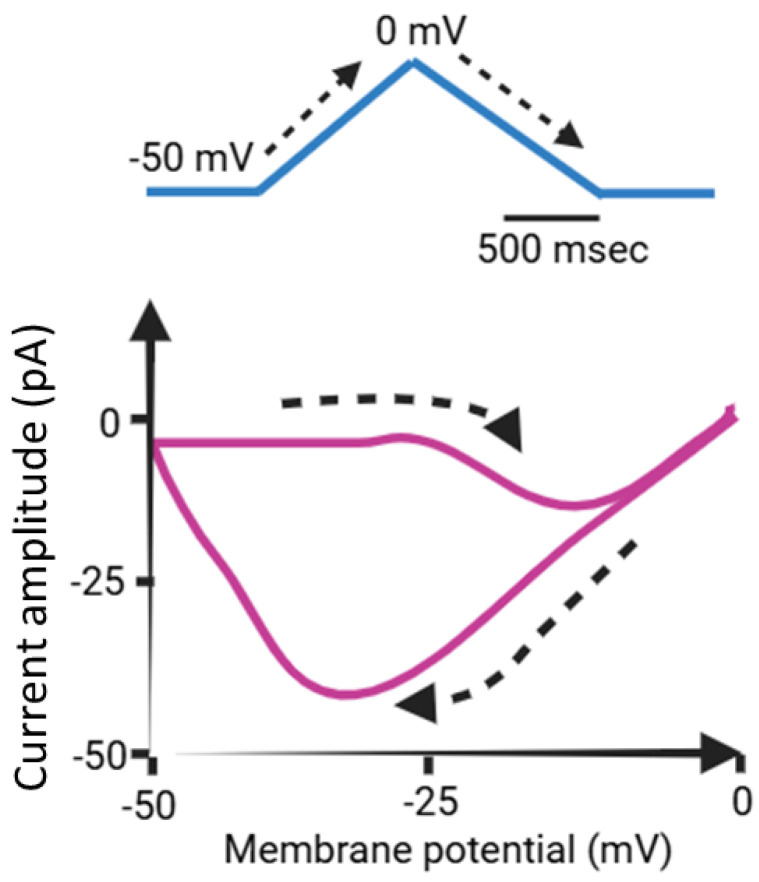
Simplified graphic representation of voltage-dependent hysteresis (Hys_(V)_) in *I*_K(M)_. The upper panel illustrates a schematic of a prolonged upright isosceles-triangular waveform, representing a double ramp voltage (V_ramp_) protocol applied over 2 s. The **lower** panel shows the relationship between voltage and whole-cell *I*_K(M)_, emphasizing the Hys_(V)_ behavior, highlighted in purple. The dashed black lines in the **upper** and **lower** panels indicate the trajectories of the applied potential and the resulting current trace over time, respectively.

**Figure 4 ijms-26-01504-f004:**
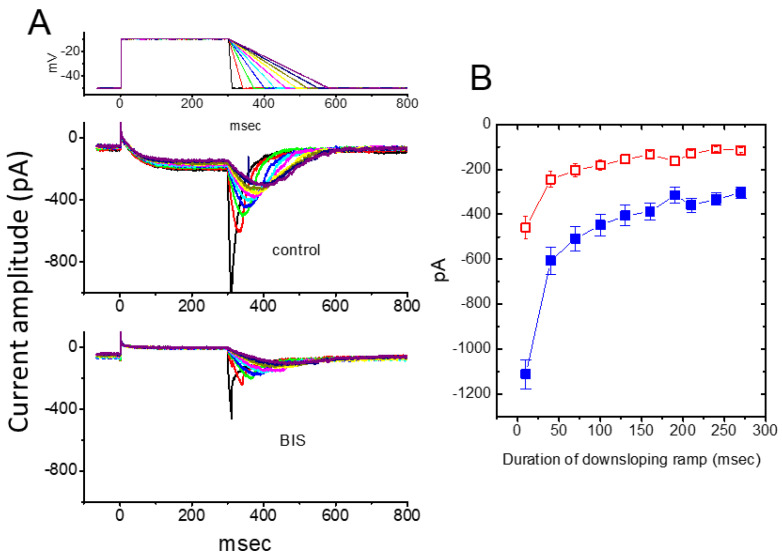
Effect of BIS on the deactivating tail *I*_K(M)_ during repolarizing phases with varying durations (109–280 ms) to simulate different repolarizing slopes of bursting action potential patterns. (**A**) Representative current traces recorded in response to the voltage protocol shown in the topmost panel, obtained in the absence (**upper**) and presence (**lower**) of 10 μM BIS. The topmost panel indicates the applied voltage protocol. The different colors in current traces correspond with those in the voltage-clamp protocols shown at the topmost panel. (**B**) Effect of BIS on the peak amplitude of deactivating *I*_K(M)_ upon return to −50 mV with different falling phase durations (mean ± SEM; n = 11 for each point). The peak amplitude indicated at each point was measured at various falling phase durations. ■: control (in the absence of BIS); □: in the presence of 3 μM BIS. This figure is adapted from So et al. [56] and published under the terms and conditions of the Creative Commons Attribution (CC BY) license.

**Figure 5 ijms-26-01504-f005:**
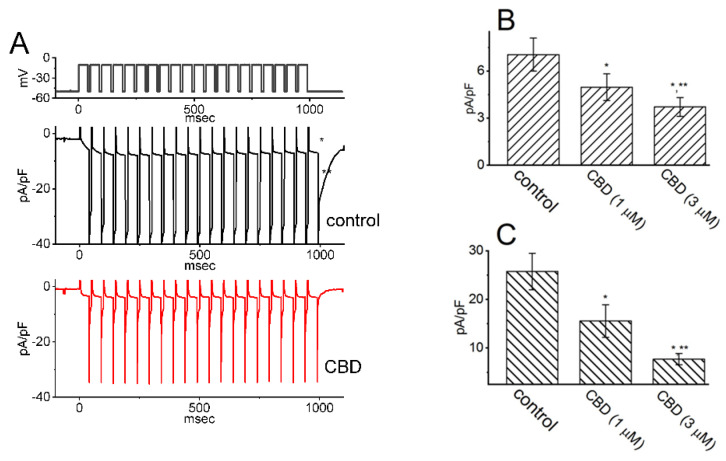
Effect of CBD on the activation of *I*_K(M)_ through pulse-train stimulation in GH_3_ cells. To perform the experiment, cells were placed in a high-K^+^, Ca^2+^-free solution, and the pulse-train stimulation protocol involved a series of 40 depolarizing pulses lasting 20 ms each, applied at −10 mV with 5 ms intervals, for a total duration of 1 s. (**A**) Representative current traces acquired during the control period (upper trace in black) and in the presence of 3 μM CBD (lower trace in red). The top portion of the figure shows the voltage-clamp protocol applied. The * symbol in the middle portion of the figure indicates the activating *I*_K(M)_, while ** represents the deactivating (or tail) component of *I*_K(M)_ obtained after returning to −50 mV. Summary bar graphs in (**B**,**C**) display the activating and deactivating densities of *I*_K(M)_, respectively, in the absence and presence of 1 or 3 μM CBD. The values are presented as mean ± SEM, with each bar representing data from seven independent experiments. The activating density of *I*_K(M)_ was measured at the end of the pulse-train stimuli from −50 to −10 mV, while the deactivating density was measured following the return to −50 mV. The * symbol indicates statistical significance when compared to the control group (*p* < 0.05), while ** denotes statistical significance when compared to the CBD (1 μM) alone group (*p* < 0.05). This figure is adapted from Liu et al. [76] and published under the terms and conditions of the Creative Commons Attribution (CC BY) license.

**Figure 6 ijms-26-01504-f006:**
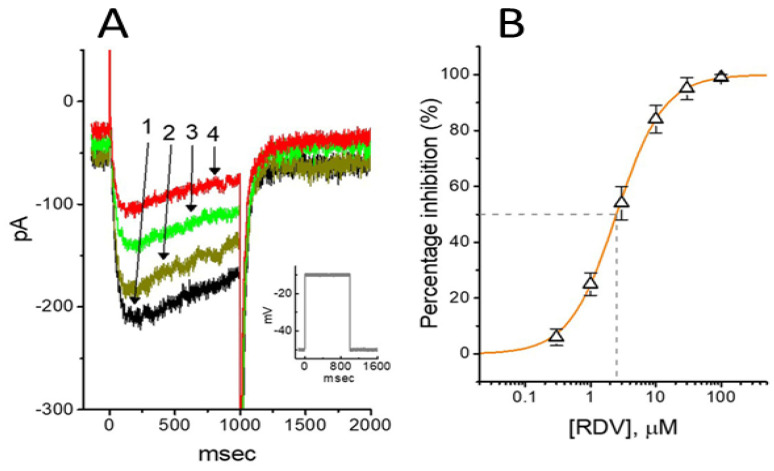
Effect of RDV on *I*_K(M)_ in GH_3_ cells. The experiments were conducted in cells immersed in high-K^+^, Ca^2+^-free solution, and the pipette used was filled with K^+^-containing solution. (**A**) Reprsentative *I*_K(M)_ traces elicited by 1 − s step depolarization from −50 to −10 mV (indicated in inset). The current trace labeled 1 is the control, and that labeled 2, 3, or 4 was obtained after the addition of 0.3 μM RDV, 1 μM RDV, or 3 μM RDV, respectively. (**B**) Concentration-dependent inhibition of RDV on *I*_K(M)_ amplitude in GH_3_ cells (mean ± SEM; n = 9). Current amplitude was measured at the end of the 1 − s depolarizing pulse. The continuous line was fitted by a Hill function. The IC_50_ value (as indicated in the vertical dashed line) needed for an RDV-induced decrease in *I*_K(M)_ was 2.5 μM. This figure is adapted from Chang et al. [101] and published under the terms and conditions of the Creative Commons Attribution (CC BY) license.

**Figure 7 ijms-26-01504-f007:**
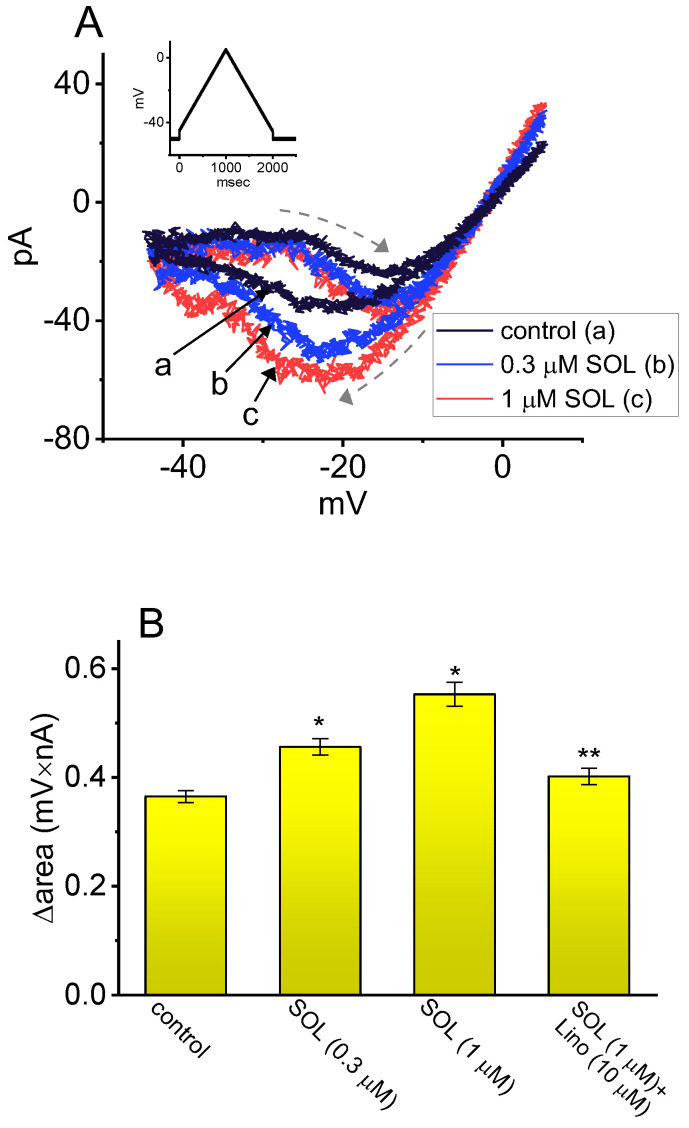
Stimulatory effect of SOL on voltage-dependent hysteresis (Hys_(V)_) of *I*_K(M)_ in GH_3_ cells. This series of experiments was conducted with an isosceles-triangular V_ramp_. (**A**) Representative current traces evoked by isosceles-triangular V_ramp_ for a duration of 2 s obtained in the control period (**a**, black) and during the exposure to 0.3 μM SOL (**b**, blue) or 1 μM SOL (**c**, red). The dashed arrows indicate the distinctive patterns of current trajectory by which time passes as V_ramp_ is applied. The voltage-clamp pulse is illustrated in inset at the left upper corner. (**B**) Hysteretic area (i.e., Δarea of *I*_K(M)_’s Hys_(V)_ obtained in control period (i.e., SOL was not present) or during exposure to SOL and SOL plus linopirdine (Lino). The area encircled by current amplitudes activated in the ascending and descending limbs at the voltages between −45 and 0 mV was calculated. Each bar indicates the mean ± SEM (n = 7 for each bar). The * symbol indicates statistical significance when compared to the control group (*p* < 0.05), while ** denotes statistical significance when compared to the SOL (0.3 μM) alone group (*p* < 0.05). This figure is adapted from Cho et al. [21] and published under the terms and conditions of the Creative Commons Attribution (CC BY) license.

**Table 1 ijms-26-01504-t001:** Two-dimensional chemical structures of drugs and compounds known to inhibit *I*_K(M)_, as detailed in this study. The data were sourced from PubChem (https://pubchem.ncbi.nlm.nih.gov/) (accessed on 9 February 2025).

Compound or Drug	Abbreviated Name	PubChem Identifier	Chemical Structure
Bisoprolol	BIS	2405	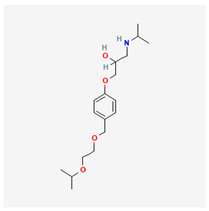
Brivaracetam	BRV	9837243	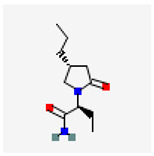
Cannabidiol *	CBD	644019	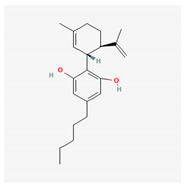
Nalbuphine	NAL	5311304	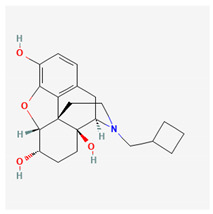
Phenobarbital	PHB	4763	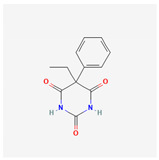
Remdesivir	RDV	121304016	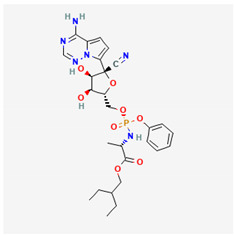

* The effect of CBD on *I*_K(M)_ is still somewhat controversial, with reports suggesting a stimulatory effect on it [23,49]. Therefore, there is still some debate on this matter.

**Table 2 ijms-26-01504-t002:** Two-dimensional chemical structures of drugs and compounds known to stimulate *I*_K(M)_, as detailed in this study. The data were sourced from PubChem (https://pubchem.ncbi.nlm.nih.gov/) (accessed on 7 February 2025).

Compound or Drug	Abbreviated Name	PubChem Identifier	Chemical Structure
Flupirtine	FLU	53276	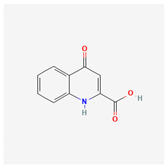
Kynurenic acid	KYNA	3845	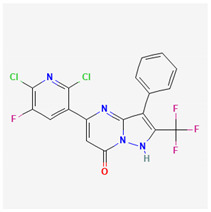
Naringenin	NGEN	439246	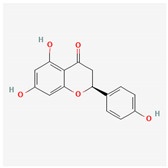
QO-58	-	51351551	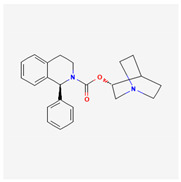
Solifenacin	SOL	154059	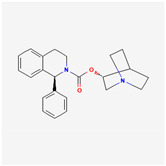

## Data Availability

The data are available upon reasonable request to the corresponding author.

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
