# Peer review of "Evaluation of Small-Molecule Candidates as Modulators of M-Type K+ Currents: Impacts on Current Amplitude, Gating, and Voltage-Dependent Hysteresis"

_ijms, 2025, doi:10.3390/ijms26041504_

Round 1
Reviewer 1 Report
Comments and Suggestions for Authors
The authors present a comprehensive overview over the physiological and pharmacological properties of M-type K+ currents (IKM) mediated by KV7.2, KV7.3, and KV7.5 channels.
Among other things, this is valuable for pharmacological research into the use of agonists or antagonists of these channels for therapeutic purposes, especially in diseases which disturbet electrical activity of neurons, like epileptic or psychiatric disorders. The manuscript also provides insights into possible side effects of drugs that stimulate or inhibit these channels.
Some formal inaccuracies should be however corrected.
P2: “Among these channels, KV7.2 and KV7.3 serve as the primary molecular components of IK(M),…”
Some remarks about KV7.5 would be helpful.
P2: “where it forms the slow delayed-rectifies K+ current (IKs)”
Better: “where it forms the slow delayed-rectifier K+ current (IKs)”
P2: “where it helps regulate the K+ balance in the cochlear …”
Better: “where it helps to regulate the K+ balance in the cochlea …”
P2: “In this review, we first describe the biophysical properties of IK(M).”
This sentence could be omitted since the information is already given in the headline.
P2: “The use of high-K+ bathing so-lution has the advantage of being convenient for distinguishing different types of voltage-gated K+ currents. These K+ currents”
What are “these currents”?
P2: “Considering the reversal potential of K+ ions is estimated to be around 0 mV, …”
Better: “Considering the reversal potential of K+ ions is around 0 mV here, “
Figure 1: “The bottom diagram represents the IK(M) trace (indicated in purple). The asterisk (*) marks the activation phase of the current, while the double asterisk (**) indicates the deactivation phase. The dashed curve illus-trates the activation and deactivation time courses of IK(M).”
That is not a trace, but a schematic depiction of this. An original current trace should be shown.
Neither asterisks nor dashed lines are shown. Please explore the horizontal black line.
Page 3: Description of the variables in the equation:
The physical quantities I and V are not explained.
Figure 2: The physical variable V is missing on the abscissa.
Figure 3: The voltage protocoll doesn't fit to the lower graph. Abscissa and ordinate are not labeled.
P6: “These results indicate that, as the duration of the duration of the downsloping Vramp in-creased, the amplitude of deactivating IK(M) decreased exponentially. “
Better: “These results indicate that, as the duration of the duration of the downsloping Vramp is in-creased, the amplitude of deactivating IK(M) decreased exponentially.
Furthermore, a different Font is used, which several times appears in the manuscript and may indicate that copy and paste from other manuscripts was used. Please correct this.
Figure 4: In A, the physical variables are missing. The legend contains different fonts. The symbols of Fig. B are incorrectly explained.
P8: “Additionally, CBD suppressed the density of both activating and deactivating IK(M) triggered by pulse-train depolarizing stimuli from -50 to -10 mV, as demonstrated in Figure 4.”
This is not shown.
P8: “The results suggest that the persistent effectiveness of CBD-induced inhibition of IK(M) even under conditions of high pulse-train stimulation “
Check this unclear sentence.
P8: “The present observations showed”
What are the present observations?
P13: “Moreover, as shown in Figure 5, previous …”
This rather refers to Figure 6.
P14: Figure 5 should be placed near the CBD paragraph.
Author Response
Dear Reviewer,
Thank you for your valuable comments and suggestions regarding our manuscript. We have carefully addressed all your comments and included our detailed responses in red font. Additionally, we have revised the manuscript accordingly to incorporate the suggested changes. The revised manuscript and the response to the reviewers are now uploaded for your consideration. We hope that the revisions meet your expectations and improve the quality of the manuscript.
Thank you again for your time and effort in reviewing our work.

Reviewer 2 Report
Comments and Suggestions for Authors
The submitted manuscript provides a timely and interesting as well as quite comprehensive summary/review of the effects of small molecule candidates and available drugs that can alter the magnitude and time course of the family of potassium currents that have been denoted M currents. Strengths of this paper as submitted include the very useful summary Tables of M current activators and inhibitors, and the illustrations of the lasting effects (denoted hysteresis) of M current activation. Against these strengths, I offer the following requirements and recommendations for changes and clarifications when the authors revise their review manuscript.
Major Comments
1. The authors choose to describe and illustrate their work in this field and the findings of others in terms of hysteresis of the M type potassium channels including references to literature depicting the hysteresis in integrated circuits. This may be useful but it completely omits what is well known concerning the subcellular and molecular basis for both the slow deactivation and the residual activation of M type potassium channels. Please review and make specific reference to the concepts and findings cited in the two papers below.
Biophysical physiology of phosphoinositide rapid dynamics and regulation in living cells. Jensen JB, Falkenburger BH, Dickson EJ, de la Cruz L, Dai G, Myeong J, Jung SR, Kruse M, Vivas O, Suh BC, Hille B. J Gen Physiol. 2022 Jun 6;154(6):e202113074. doi: 10.1085/jgp.202113074. Epub 2022 May 18. PMID: 35583815
Phosphoinositides regulate ion channels. Hille B, Dickson EJ, Kruse M, Vivas O, Suh BC. Biochim Biophys Acta. 2015 Jun;1851(6):844-56. doi: 10.1016/j.bbalip.2014.09.010. Epub 2014 Sep 18. PMID: 25241941
2. Your referencing is certainly extensive but my preference would be for your article to start with a more clear and correct description of the discovery of the M current. Please refer to the 1980 Nature paper cited below and a related review article.
Muscarinic suppression of a novel voltage-sensitive K+ current in a vertebrate neurone. Brown DA, Adams PR. Nature. 1980 Feb 14;283(5748):673-6. doi: 10.1038/283673a0. PMID: 6965523
Pharmacological inhibition of the M-current. Adams PR, Brown DA, Constanti A. J Physiol. 1982 Nov;332:223-62. doi: 10.1113/jphysiol.1982.sp014411. PMID: 6760380
3. Of less importance but of interest, your description of the small molecule and drug effects on the putative steady-state activation curve for the M current should perhaps make reference to the insights provided by the recent work and previous publications the Elinder group. The material that I have in mind can be found in the recent paper that is cited below.
Carboxyl-group compounds activate voltage-gated potassium channels via a distinct mechanism. Rönnelid O, Elinder F. J Gen Physiol. 2024 Jul 1;156(7):e202313516. doi: 10.1085/jgp.202313516. Epub 2024 Jun 4. PMID: 38832889
Minor Comments
Your paper is logically presented and quite well illustrated. There are however, key phrases and some paragraphs that require editing. Examples include but are not limited to the following.
i) The title - The wording 'Druggability of small-molecule...' is unconventional. I would suggest 'Evaluation of small molecule candidates as modulators of...'
ii) Page 2, paragraph 2 - Your description of Kv7.1 in the heart is incomplete and could be misleading. It is likely that much of the Kv7.1 in heart is in fact in intracardiac neurons; and I am unaware that this potassium channel could be accurately described as 'highly expressed' anywhere in the myocardium.
iii) Page 4, paragraph 2 – Here and elsewhere in this interesting review, you mention and make use of ‘differences in free energy associated with the gaiting of’. This concept and its functional importance is not introduced sufficiently or adequately explained. This is particularly true when one recognizes the importance of phosphoinositide metabolism as the primary second messenger.
iv) Page 5, paragraph 2 – In the sentence beginning, ‘Consequently’, I would suggest that the main influence of the M current in small cells is its modulation of the resting potential and associated input resistance.
v) Page 6, paragraph 3 – The sentence beginning, ‘However’, seems incomplete and in the next paragraph, what do you mean by ‘the falling phase of burst firing’, and ‘as the duration of the duration’ actually mean?
vi) Page 8, under the Cannabidiol section, I find paragraphs 2 and 3 to be unclear. I would recommend rewriting this important material.
vii) Page 9, under the Phenobarbital section, I find the material and argument to be unconvincing. You cite these drug effects as being produced by 100 – 300 micromolar doses. These changes are extremely unlikely to be due to selective drug effects.
viii) Page 10, in the Remdesivir section, I would recommend inserting a Figure of original data. In my view this material may be the most interesting and original effects that you describe.
ix) Page 11, paragraphs 1 and 2 – Understanding these results requires the reader to have some concept of what a ‘resurgent K current’ actually is. Please provide a brief introduction.
x) Pages 12 and 13, your description of the strength of the hysteresis relationship concerns me given the extensive work of the Hille group that is cited above.
Comments on the Quality of English Language
This is a well planned, quite comprehensive and clearly presented review manuscript. However, there are existing substantial concepts and primary publications that will need to be cited; and editorial work that should be done to improve clarity and remove unconventional phrases.
Author Response

(The authors gave the same response as above.)

Round 2
Reviewer 2 Report
Comments and Suggestions for Authors
Thank you for considering and responding to my questions and comments. Judged as a whole, these changes and the new Figure clarify and improve your manuscript. However, one of your responses is inadequate and this may be because my original review failed to make a point with sufficient clarity. It is clear that the previous work from the Hille group cited under my major comment #1 provide an important and, to me, compelling mechanistic interpretation based on phosphoinositide metabolism and spatial localization of the phenomenon that you denote 'hysteresis'. You need to further revise the manuscript and make a much more meaningful response than 'adding an additional sentence'. This same concern applies in the extent and the wording of your response to my page 4, paragraph 2 comment.
Author Response
Dear Reviewer,
Thank you for your valuable comments and suggestions regarding our manuscript. We have carefully addressed all your comments and included our detailed responses in red font. Additionally, we have revised the manuscript accordingly to incorporate the suggested changes. The revised manuscript and the response to the reviewers are now uploaded for your consideration. We hope that the revisions meet your expectations and improve the quality of the manuscript.

Round 3
Reviewer 2 Report
Comments and Suggestions for Authors
Thank you for considering and responding effectively to my request for a more specific citation of the body of work from the Hille lab that is a direct and important precedent to your presentation and interpretation of the hysteresis exhibited by M currents.